

# Using neural networks for near-real-time aerosol retrievals from OMPS Limb Profiler measurements

Michael D. Himes[1,2], Ghassan Taha[3,1], Daniel Kahn[4], Tong Zhu[4], and Natalya A. Kramarova[1]

[1]NASA Goddard Space Flight Center, Greenbelt, MD, USA
[2]NASA Postdoctoral Program Fellow
[3]Morgan State University, Baltimore, MD, USA
[4]Science Systems and Applications, Inc., Lanham, MD, USA

**Correspondence:** Michael D. Himes (michael.d.himes@nasa.gov)

**Abstract.** Among aerosol characterization methods, limb scattering measurements provide both near-global coverage and information about how aerosol is vertically distributed through the atmosphere. Near-real-time characterization of aerosols produced by volcanic eruptions is particularly important for aviation safety, but the radiative transfer modeling of scattering processes performed by traditional retrieval methods are too computationally expensive for near-real-time applications without

simplifying assumptions. Here we present a near-real-time approach based on neural networks (NNs) for aerosol retrievals from the Ozone Mapping and Profiler Suite's Limb Profiler (OMPS LP) instrument aboard the Suomi National Polar-orbiting Partnership satellite. We find it is at least 60 times faster than the current operational code and on average achieves agreement within 20% at most altitudes and latitudes with sensitivity and non-negligible aerosol abundances. We also apply our trained NNs to measurements of the recent Shiveluch and Ruang eruptions from NOAA-21's OMPS LP and find results consistent

with the operational retrieval algorithm, indicating our methodology generalizes to future iterations of the same instrument without re-training the NNs.

## 1 Introduction

Stratospheric aerosols play a key role in the atmospheric radiation budget. Negative radiative forcing due to background sulfate aerosols can slow the rate of increase in surface temperature (Solomon et al., 2011), and the increased aerosol load from

eruptions like El Chichón, Pinatubo, and Hunga Tonga-Hunga Ha'apai (HTHH) can lead to periods of increased outgoing shortwave radiation and cooler temperatures in the lower troposphere (Santer et al., 2014; Schoeberl et al., 2023). Major wildfires can trigger pyrocumulonimbus (pyroCb) events and cause stratospheric warming (Christian et al., 2019; Das et al., 2021). Characterizing these stratospheric aerosols therefore constrains climate and chemical modeling.

    Stratospheric aerosols can be characterized through in-situ, ground-based, and space-based measurements (see Section 4 of

Kremser et al., 2016 for a detailed review). In-situ (e.g., Junge et al., 1961; Hofmann et al., 1975; Jonsson et al., 1995; Brock et al., 2000; Hermann and Wiedensohler, 2001; Borrmann et al., 2010) and ground-based lidar (e.g., Fiocco and Grams, 1964; Poole and McCormick, 1988; Osborn et al., 1995; Barnes and Hofmann, 1997; Jäger, 2005; Chouza et al., 2020) measurements can provide high vertical resolution at the expense of limited spatial coverage. Space-based measurements, on the other hand,



offer much broader spatial coverage. Solar occultation measurements (e.g., McCormick et al., 1982; Chu et al., 1989; Hervig
et al., 1996; Hayashida et al., 2000; Lumpe et al., 2002; Thomason and Taha, 2003; McElroy et al., 2007) allow for the highest
precision and accuracy among space-based methods and provide information about the vertical distribution of aerosols, but it is
only possible during sunrise/sunset and thus has limited spatial and temporal coverage. Space-based nadir lidar measurements
(e.g., Vernier et al., 2011; Kar et al., 2019) provide better spatial and temporal coverage, but the measurements typically have
low signal-to-noise ratios for stratospheric aerosol, limiting their precision. Additionally, they measure polarization and aerosol
backscatter coefficient rather than aerosol extinction, and they have more significant power requirements compared to other
space-based methods. Limb-scattering (LS) measurements (e.g., Llewellyn et al., 2004; Rault and Taha, 2007; Taha et al.,
2011; Loughman et al., 2018; von Savigny et al., 2015) serve as a middle ground among space-based methods, providing near-
global coverage of the vertical distribution of aerosols with lesser cost and power requirements compared to other measurement
techniques.

The Ozone Mapping and Profiler Suite (OMPS) Limb Profiler (LP) instrument (Flynn et al., 2006) first launched aboard the
Suomi National Polar-orbiting Partnership (SNPP) satellite in 2011 and subsequently aboard the NOAA-21 satellite in 2022.
OMPS LP measures LS radiances between 290–1000 nm at a roughly 1 km vertical resolution through three slits; the lines of
sight for the left and right slits are separated by approximately 4.25° with respect to the center slit. Measurements are made of
the sunlit limb 180 times per orbit for 14–15 orbits per day.

Taha et al. (2022) introduced version 2.1 of the OMPS LP aerosol retrieval algorithm, which improved on the convergence
criteria used in version 2.0 (Taha et al., 2021). It uses the Gauss-Siedel LS radiative transfer model (Loughman et al., 2004)
to retrieve the aerosol extinction coefficient up to 38.5 km at wavelengths of 510, 600, 675, 745, 869, and 997 nm. Each
wavelength is retrieved independently. For more details, see the aforementioned references.

Near-real-time (NRT) characterization of aerosols is particularly important for informing aviation flight paths in the vicinity
of a recent volcanic eruption as well as providing advance notice to coordinate ground-based or in-situ follow-up measurements.
However, while aerosol retrieval algorithms like that of Taha et al. (2022) provide a consistent, physics-based result, their
runtime is prohibitive to NRT applications unless compromises are made, such as retrieving at a reduced number of wavelengths
and/or slits.

Deep learning, a subfield of machine learning that focuses on neural networks (NNs; Goodfellow et al., 2016), offers a
potential solution to this problem. Given a set of corresponding input-output pairs, NNs can learn to approximate complex,
nonlinear relationships between inputs and outputs via a sequence of weighted transformations, without knowledge about the
underlying physical process that produced that connect inputs and outputs. This can be particularly useful to approximate a
computationally expensive algorithm (e.g., Baydin et al., 2019; Cranmer et al., 2020; Munk et al., 2020; Kasim et al., 2021;
Himes et al., 2022), such as aerosol retrievals from limb-scattered radiances. The trained NN model introduces some error in
the outputs, but provided that the error is small enough for the desired applications, the resulting NN can be used in place
of the computationally expensive algorithm. Werner et al. (2023) recently demonstrated this for NRT products from the Aura
Microwave Limb Sounder; they found that NNs trained on the level 2 products generally outperformed the optimal estimation-
based NRT product.



**Table 1.** Date ranges of OMPS LP data considered.

| Date ranges | Notable events |
| --- | --- |
| October 2013 – October 2014 | Quiescent period |
| May 2015 – November 2016 | Calbuco |
| August 2017 – June 2018 | Pacific Northwest wildfires |
| June 2019 – June 2021 | Raikoke, Ulawun, Australian wildfires |
| January – December 2022 | Hunga Tonga-Hunga Ha'apai |

**Notes.** We select 10% of the above dates, totalling 241 days. We specifically exclude the dates where OMPS LP initially measured the plume from the Hunga Tonga-Hunga Ha'apai eruption from the training data so they can be used as test cases.

Here we present a NRT data product for retrieved aerosol extinction and related parameters from OMPS LP measurements based on NNs. In Section 2 we detail our methodology. We present comparisons with the existing aerosol data product and discuss our results in Section 3. Finally, we present our conclusions in Section 4.

## 2 Methods

In this study, we seek to train NN models that can accurately retrieve the aerosol extinction coefficient from SNPP OMPS LP measurements.

### 2.1 Data selection

Much of the OMPS LP record measures low stratospheric aerosol loading; elevated stratospheric aerosol from major volcanic and wildfire events can last for months to years, depending on the amount of aerosol produced. We utilize measurements during specific periods between October 2013 and December 2022 (Table 1). The periods were determined from Figure 10 of Peterson et al. (2021) to capture a quiescent baseline as well as major volcanic and wildfire events that have occurred during SNPP's lifetime. We randomly select 10% of these dates, totalling 1.24 million profiles spread throughout 241 days.

Utilizing these data, we prepare the input-output pairs to use during NN training. For each case, the inputs are comprised of

- the version 2.6 OMPS LP gridded sun-normalized measured radiances (Jaross, 2023) between 0.5–40.5 km, with altitude normalization at 38.5 km,

- atmospheric pressure and temperature from the NASA Global Earth Observing System Forward Processing for Instrument Teams product (GEOS FP-IT; Lucchesi, 2015),

- single-scattering angle (calculated from quantities available in L1G),

- solar zenith angle, and





    – spacecraft-centered solar azimuth angle.

These parameters were selected given their use in the operational aerosol retrieval algorithm and availability in the Level 1

gridded radiance version 2.6 data product and the associated ancillary files. These inputs correspond to the aerosol extinction coefficient reported in the OMPS LP aerosol retrieval version 2.1 data product (V2.1; Taha et al., 2022). For the input radiances and retrieved extinction coefficients, we consider wavelengths of 510, 600, 675, 745, 869, and 997 nm. At altitudes where the aerosol extinction coefficient was not retrieved, we assume a value of $10^{-8}$, which is effectively treated as zero aerosol extinction in V2.1. This replacement is required as the NNs cannot handle missing data. During post-processing, any values

predicted by the NN to be less than $10^{-8}$ are replaced with a fill value of -999, for consistency with the V2.1 data product. When preparing the input-output pairs, slit position is not included as a parameter or dimension. The resulting trained NN can therefore be applied to measurements from any of the three slits, thereby minimizing the up-front computational costs of training the NN and enabling retrievals in situations where only 1 or 2 slits have valid measurements.

    We highlight that the current OMPS LP aerosol retrieval algorithm (V2.1) uses the version 2.5 radiances, while here we

use the version 2.6 radiances as input because it is the only version of radiances processed in NRT. This complicates the relationship between inputs and outputs, as the version 2.6 radiances use different tangent height and stray light corrections. These correction methods introduce differences in the retrieved aerosol extinction coefficient, and so our input-output pairs do not have a perfect one-to-one relation. However, no OMPS LP aerosol retrieval version uses the version 2.6 radiance data as input, and retroactively processing the OMPS LP record using that version is computationally prohibitive. Since the

differences in correction methods are consistent, our methodology ignores them and assumes that the NN will learn to perform the transformation from version 2.6 radiances to V2.1 aerosol extinction coefficients. To confirm the validity of this assumption, we processed a limited number of retrievals on L1G version 2.6 and trained NNs as described below.

## 2.2   Neural network training and evaluation

These data are split into training, validation, and test sets. We enforce that the days which contain extrema for each input/output

parameter must be within the training set. The remaining days are randomly split among the training, validation, and test subsets, in a proportion of approximately 70%, 20%, and 10%, respectively. To pre-process these data and train the NNs, we use the open-source Python package, MARGE (Himes et al., 2022), which is built on TensorFlow (Abadi et al., 2016). We normalize these data by taking the base-10 logarithm of the radiances, pressure, and aerosol extinction, then scaling each input and output parameter to be within the closed interval [-1, 1] based on their training set extrema.

To determine NN architectures which are well suited to solve the problem of interest, we perform a Bayesian hyperparameter optimization. We consider models with 2–5 convolutional and/or fully-connected hidden layers. The convolutional layers are allowed to have $2^x$ feature maps with $6 \leq x \leq 8$, while the fully-connected layers are allowed to have $2^y$ nodes with $6 \leq y \leq 10$. For each hidden layer, we consider rectified linear unit (ReLU), exponential linear unit (ELU), leaky ReLU (LReLU), sigmoid, and hyperbolic tangent (tanh) activation functions. For ELU and leaky ReLU activation functions, we allow their free parameter

to vary between 0.01 and 0.6. For all models, the input and output layers have linear activations, and the output layer is a fully-



connected layer with a number of nodes based on the target outputs (6 wavelengths × 41 altitudes). During the Bayesian optimization, each considered model is optimized according to the validation mean-squared-error (MSE) loss and trained for 60 epochs using the Adam optimizer, a batch size of 256, and a cyclical learning rate that linearly cycles between $10^{-5}$–$10^{-3}$ over 12 epochs (Smith, 2015; Himes et al., 2022).

Based on this optimization, we selected an architecture with hidden layers of Conv3D(256)—LReLU(0.0733)—FC(1024)—tanh—FC(512)—ELU(0.2944)—FC(1024)—ReLU, where Conv3D($m$) denotes a 3D convolutional layer with $m$ feature maps, LReLU($s$) denotes a leaky ReLU activation with slope parameter $s$, FC($n$) denotes a fully connected layer with $n$ nodes, tanh denotes a hyperbolic tangent activation, and ELU($\alpha$) denotes an ELU activation with scaling parameter $\alpha$. We note that while different architectures result in variations in model performance, many different architectures can achieve similar

performance; the training data, its preprocessing, and the training methodology had a more significant effect on the resulting model accuracy.

Using the selected architecture, we train two separate NNs, as we found poor performance at 510, 600, and 675 nm in the southern hemisphere, where OMPS LP observes backscattered radiation, when using a single NN. For each NN, we determine a cyclical learning rate range based on the range test described in Himes et al. (2022). Unlike in Himes et al. (2022), here

we utilize real data during NN training; our problem requires that the minimum learning rate must be increased above that determined via the range test described in Himes et al. (2022) to avoid overfitting the validation data. We train one NN using a cyclical learning rate varying between $3 \times 10^{-6} - 10^{-3}$ and a MSE loss; it is used for the northern hemisphere at 675 and 745 nm and for all latitudes at 869 and 997 nm. We train the second NN with a cyclical learning rate over the range of $10^{-5}$ $- 2 \times 10^{-3}$ and a custom loss function which minimizes the maximum MSE across wavelength channels; it is used for all

latitudes at 510 and 600 nm and in the southern hemisphere for 675 and 745 nm. We allow each NN to train until engaging early stopping after a patience of 60 epochs.

To thoroughly evaluate the NN performance, we compare the retrieved extinction coefficient and the stratospheric aerosol optical depth with the current OMPS LP aerosol data product over January 2019 – December 2023 and compute a mean percent error with respect to altitude and latitude. We also test the NN predictions for the 2022 HTHH eruption to ensure the models can be applied to extreme cases beyond those seen in the training set as well as the 2023 Shiveluch and 2024 Ruang eruptions to ensure the models are accurate when applied outside of the range of years considered in the training data set. We also utilize the 2024 Ruang eruptions to demonstrate how visualizations from our NRT product can accurately track the transport of a volcanic plume. Finally, given that OMPS LP is planned to launch on multiple additional satellites over the next decade, we briefly consider the NN performance when applied to selected orbits from NOAA-21's OMPS LP that measured the recent Shiveluch

and Ruang eruptions to investigate the applicability of our SNPP NRT approach to OMPS LP measurements in general.





## 3 Results and Discussion

### 3.1 Comparison with OMPS LP Version 2.1 aerosol data product

Figure 1 summarizes the difference in zonal means between the NRT and V2.1 aerosol data products over the period of January 2019 – December 2023. This period is marked by a series of volcanic eruptions and massive wildfires that injected large
amounts of aerosols into the stratosphere in both hemispheres. Errors are generally within 20% in regions with non-negligible aerosol extinction. The notable exceptions to this are lower altitudes in the southern hemisphere at the shorter wavelengths and in the troposphere, where the percent errors increase to $> 50\%$. The former regime generally has a small aerosol scattering index, which causes the retrieval algorithm to struggle to provide consistent results (Taha et al., 2021); consequently, the NNs also struggle in this regime. The large differences in the troposphere are mostly caused by cloud interference, which may cause
inaccurate retrievals for both algorithms.

Figure 2 and Figure 3 show the stratospheric aerosol optical depth (sAOD) and aerosol extinction at 20.5 km, respectively, at 997 nm between January 2019 and October 2023 for the V2.1 and NRT aerosol data products. The NRT results generally agree with V2.1, capturing increases in aerosol loading due to both eruptions and wildfires during this period. For example, aerosol due to the Raikoke eruption and northern hemisphere wildfires in 2019 are clearly visible in Figure 2; the NRT product correctly
infers the decrease in sAOD over time as well as the transport southward into the mid-latitudes. Similarly, the increased sAOD throughout 2022 associated with the HTHH eruption agrees both in the initial months after the eruption as well as later in the year when the aerosol was transported toward the southern pole. Most errors are within 20%, consistent with Figure 1. Some periods have larger percent errors, which are mostly associated with small sAOD and extinction values, such as the period leading up to the Ulawun eruption in 2019.

The HTHH eruption is the only event in this five-year period where the NRT product consistently underpredicts the extinction at 20.5 km at 997 nm (Figure 3) compared to V2.1. The NRT results show a bias up to -50% between May – September 2022 and November 2022 – January 2023; the latter period corresponds to transport into the northern hemisphere. This is likely attributable to the extreme nature of the event. Despite this, the NRT predictions for HTHH are qualitatively consistent with V2.1, and errors are generally within 20%.

Considering results for individual orbits, the NRT results generally agree with V2.1, capturing similar structures and magnitudes across the orbit. Figure 4 compares the V2.1 and NRT results at 997 nm for the OMPS LP orbit on 16 January 2022 that first measured HTHH's plume. This orbit features aerosol extinctions that are more extreme than those included in the training data set. Though it slightly overestimates the extinction below the plume and below the cloudtops in the southern hemisphere, the extinction values near the top of the plume are consistent and the plume height agrees well. Figure 5 shows the V2.1 and
NRT results at 997 nm for the center slit of the OMPS LP orbit that measured Shiveluch shortly after the eruption on 11 April 2023. While a relatively small eruption, Shiveluch highlights the importance of NRT aerosol retrievals: numerous flights to and from Alaska were canceled due to volcanic ash at airplane cruising altitudes. Despite being outside of the temporal range of data that trained the NNs, the NRT approach agrees with V2.1.





Another example is the Ruang volcano in North Sulawesi, Indonesia that erupted on 16 April 2024 and continued intermit-
tently through 23 April. The largest emissions occurred on 17 April, and plumes reportedly reached up to almost 20 km and
generally drifted west of the island (Global Volcanism Program, 2024). Late on 29 April, Ruang began erupting again, with
OMPS LP measuring the plume reaching up to 23.5 km. Figure 6 shows the average retrieved extinction coefficient between
19.5–21.5 km at 997 nm for the operational and NN-based algorithms for all orbits on 1 May. Aerosol from the first eruption

**Figure 1.** Plot of the mean differences in percent between OMPS LP NRT and V2.1 aerosol extinction profiles for the center slit at (a) 510,
(b) 600, (c) 675, (d) 745, (e) 869, and (f) 997 nm, zonally averaged at 5° latitudes for the period between January 2019 – December 2023.
The dashed line is the tropopause altitude. Contours are shown for ±20% errors. Errors < −50% are shown in white; this occurs where
aerosol extinction is typically negligible (upper altitudes) and where the aerosol retrieval algorithm struggles to provide consistent results
(lower altitudes in the southern hemisphere at shorter wavelengths; Taha et al., 2021).



**Figure 2.** (a) OMPS LP V2.1 daily stratospheric aerosol optical depth (sAOD) at 997 nm for the center slit, calculated in 5 degree latitude bands. (b) The same but for NRT. (c) The daily zonal differences between (a) and (b) in percent. Black markers indicate volcanic eruption or pyroCb smoke plumes.





**Figure 3.** (a) OMPS LP V2.1 daily 20.5 km retrieved aerosol extinction at 997 nm for the center slit, calculated in 5 degree latitude bands. (b) The same but for NRT. (c) The daily zonal differences between (a) and (b) in percent.





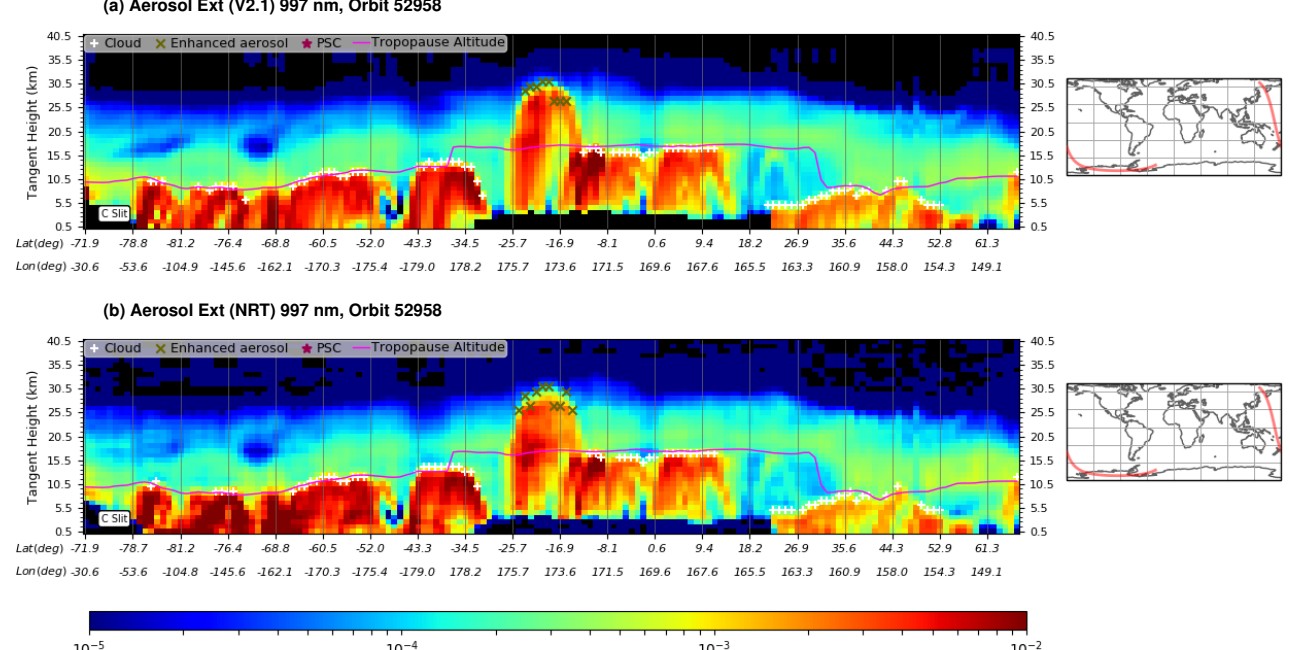

**Figure 4.** (a) OMPS LP V2.1 retrieved aerosol extinction at 997 nm for the center slit of orbit 52958 on 16 January 2022, which measured shortly after Hunga Tonga-Hunga Ha'api's eruption. (b) The same but for NRT. The plume is visible around 20°S latitude, reaching roughly 30 km in altitude.

can be seen stretching from eastern Africa to the eastern Pacific near the coast of South America, while aerosol from the second
eruption can be seen over Borneo and the Indian Ocean. We also processed the orbits between 16 April–13 May to produce a time-lapse animation of the average retrieved aerosol extinction between 19.5–21.5 km; see the video supplement (Himes et al., 2024b). This visualization represents one use case of our NRT product, enabling the tracking of volcanic plumes and (in combination with wind fields) predicting their future transport, e.g., to coordinate balloon-borne measurements or inform aviation flight paths.

The NRT approach requires just over 2 minutes to process a single OMPS LP orbit on our hardware, achieving a ∼60× speedup compared to V2.1. This significantly reduces the delay between obtaining and processing the data, which is particularly important for aviation safety in the wake of a volcanic eruption. Much of the NRT processing time is spent loading the NN models into memory; each model spends just 3–4 seconds predicting the aerosol extinction per slit. Thus, processing a range of orbits where the NN models can remain loaded in memory can substantially increase this speedup factor (e.g., processing
10 orbits would result in approximately 400× less computational time). When experimenting with changes to the radiative transfer-based aerosol retrieval algorithm, our methodology can therefore significantly reduce the computational resources required to determine how such changes would affect the mission's complete record.





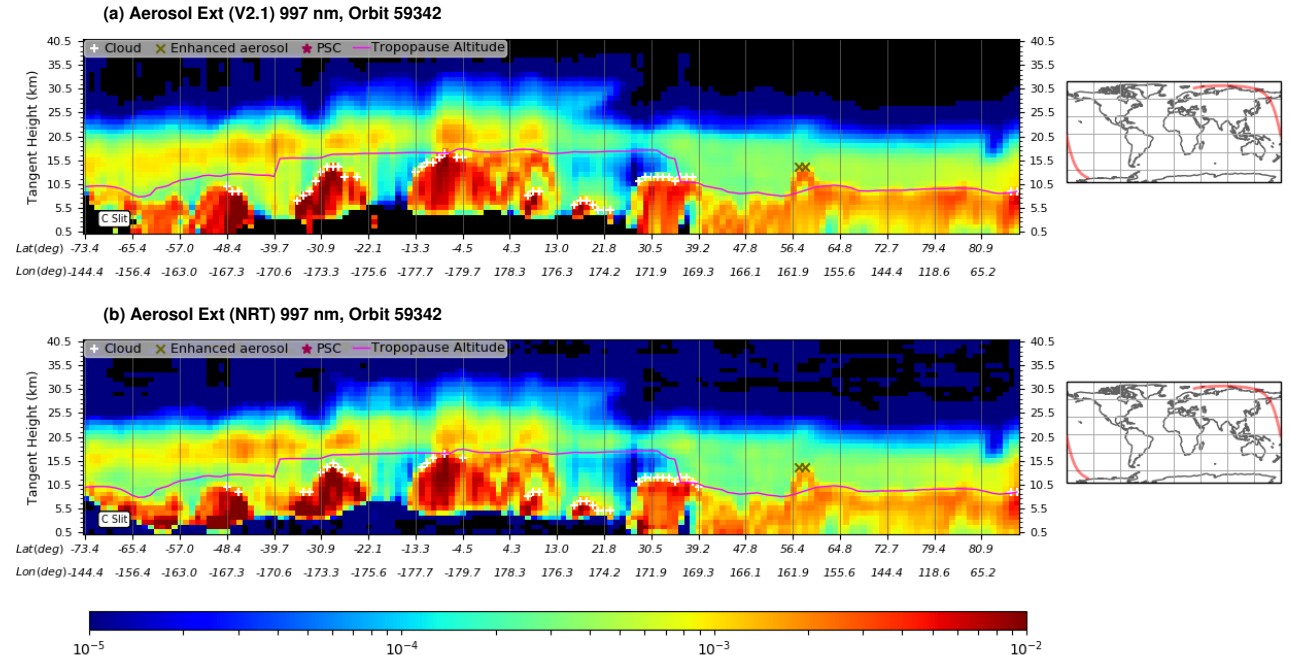

**Figure 5.** (a) OMPS LP V2.1 retrieved aerosol extinction at 997 nm for the center slit of orbit 59342 on 11 April 2023, which measured shortly after Shiveluch's eruption. (b) The same but for NRT. The plume appears above the tropopause at approximately 57°N latitude.

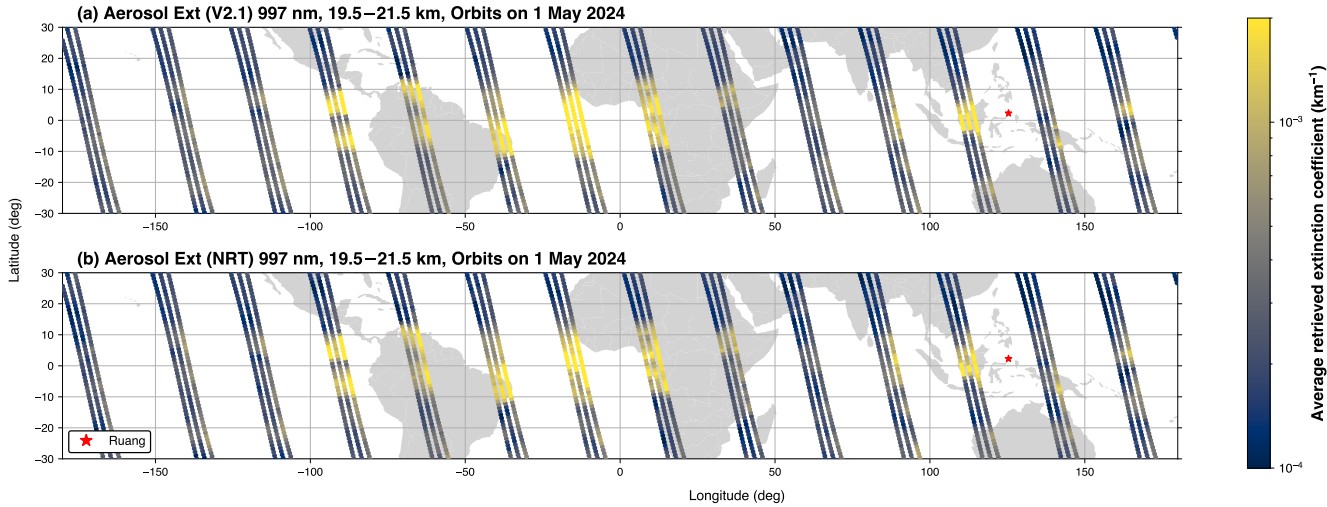

**Figure 6.** (a) V2.1 average retrieved aerosol extinction coefficient between 19.5–21.5 km at 997 nm for 1 May 2024, which shows aerosol from Ruang's recent eruptions stretching from Borneo to the eastern Pacific Ocean. (b) The same but using the NRT algorithm. Individual points are colored by the mean extinction coefficient value at each measurement location.



When developing this data product, we originally attempted to train a single NN to retrieve aerosol extinction for the entire orbit by minimizing the MSE. However, we found that this led to the NN only performing accurately in the northern hemisphere and longer wavelengths, with strong biases in the southern hemisphere and shorter wavelengths. This is likely due to a combination of increased Rayleigh scattering at shorter wavelengths as well as OMPS LP's Sun-synchronous orbit. As OMPS LP ascends from the south pole towards the north pole, the single-scattering angle changes throughout the orbit, resulting in measurements of backscattering in the southern hemisphere and forward-scattering in the northern hemisphere. This leads to a weaker aerosol scattering signal in the southern hemisphere, which can be on the order of or smaller than the Rayleigh scattering signal. To address this, we considered custom loss functions; we found that separately calculating the MSE per wavelength channel and minimizing the maximum MSE across wavelengths significantly improved performance in the southern hemisphere and shorter wavelengths, but the northern hemisphere and longer wavelengths generally performed worse compared to just minimizing the MSE. Consequently, we chose to use a combination of those models to achieve acceptable performance in both hemispheres at all wavelengths.

We also considered a more traditional surrogate modeling approach by applying the aerosol retrieval code to the L1G version 2.6 radiances for the same 241 selected days and training the NNs on these data; it achieved similar performance to that presented here (not shown). This confirms our approach's implicit assumption that the NNs can learn to handle the minor differences in corrections applied to the radiances between versions 2.5 and 2.6, circumventing the need to reprocess the aerosol retrievals after future updates to the radiance data product.

## 3.2 Limitations

Our NRT approach is subject to two types of limitations: those inherent in the operational retrieval code that produced the training data, and those specific to the NN approach used here. The operational retrieval code generally struggles to provide consistent results at high altitudes where aerosol abundances are small and in the southern hemisphere due to the afore-mentioned weak aerosol backscattering signal and increased scattering at shorter wavelengths. It also struggles when aerosol reaches the normalization altitude of 38.5 km, which happened with HTHH (Taha et al., 2022). Readers interested in more details on these limitations are directed to Taha et al. (2021, particularly Figure 7) and Taha et al. (2022). In the former case, the NRT algorithm does not yield results consistent with V2.1; rather than produce artifacts like V2.1, it tends to predict small aerosol values. In the latter case, the NRT algorithm produces artifacts in the retrieved extinction not seen in V2.1, but this situation can be identified by the reported enhanced aerosol altitude occurring at or above the normalization altitude.

The most significant NN-specific limitations are related to the input data. Our methodology relies on a diverse data set that samples low, moderate, and extreme aerosol loading in order to train NNs to correctly interpret the radiances under these conditions. These trained NNs are only applicable to OMPS LP; they cannot be reliably applied to data from any other instrument. As future improvements are made to the gridded radiances data product, these NNs may need to be re-trained. Similarly, future improvements to the aerosol retrieval code may necessitate reprocessing the selected dates used to train the NNs, then re-training the NNs to predict those data. However, these situations may require only retraining the weights



associated with the relevant layer (transfer learning), reducing the computational cost of training updated models. Future work should explore this possibility in detail.

More broadly, these NNs are limited by the extrema encountered in the training data. This limits the applicability of this methodology to recently launched instruments due to a lack of diverse data records that sample multiple major eruptions and

wildfire events in both hemispheres. While the HTHH case presented above demonstrates that the NNs can be applicable to more extreme radiances and extinctions than those encountered during training, this does not necessarily hold true for all anomalous events nor the other input parameters.

## 3.3 Application to NOAA-21 OMPS LP

Our methodology is challenged when applied to a new instrument due to the short data record that has likely not measured

sufficiently diverse conditions to train generalized NNs. NOAA-21 was launched in November 2022; the OMPS LP instrument onboard has been operational since February 2023 and thus represents an ideal test case to determine whether the SNPP NRT model can be reliably applied to NOAA-21 data, thereby circumventing the data record limitation and enabling our method to

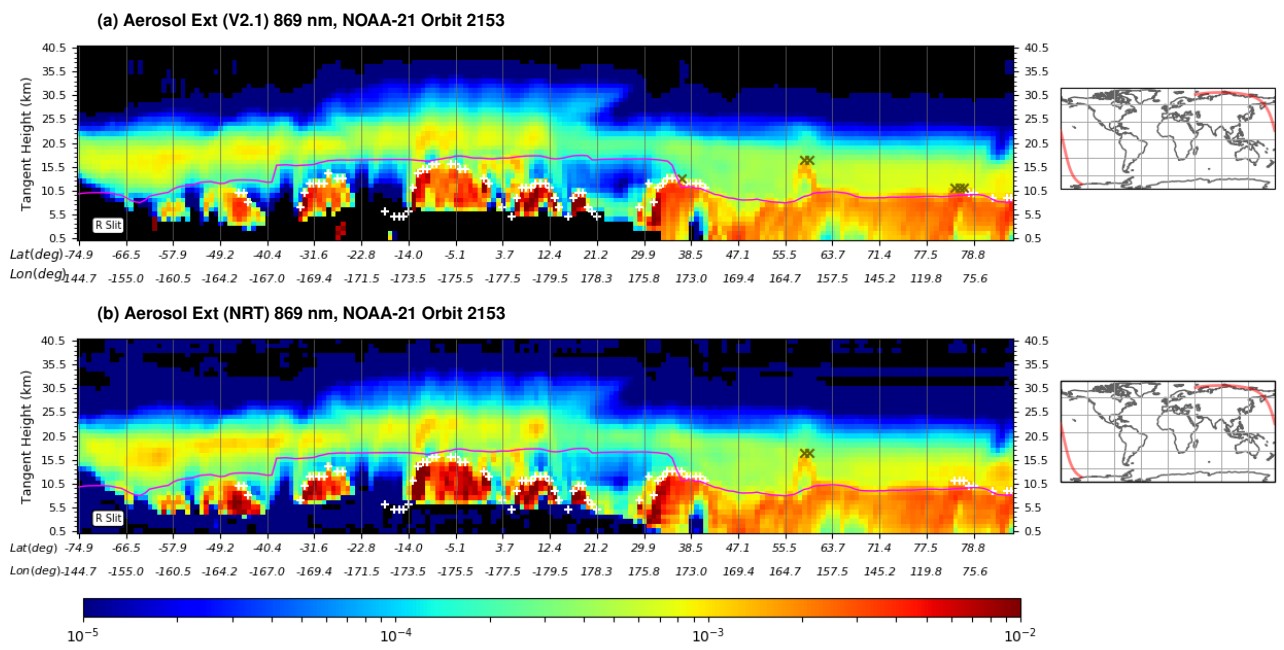

**Figure 7.** (a) NOAA-21 OMPS LP retrieved aerosol extinction at 869 nm for the right slit of orbit 2153 on 11 April 2023, which measured shortly after Shiveluch's eruption. (b) The same but using the SNPP NRT algorithm. The plume appears above the tropopause at approximately 57°N latitude.





be applied to new instruments in situations where our NRT approach has already been applied to an earlier version of the same instrument.

When applying the NN models trained using SNPP retrievals to select NOAA-21 orbits, we find that they agree qualitatively and quantitatively with the operational results, with biases that are broadly in agreement with Figure 1. For more detailed comparisons, we present two cases. Figure 7 shows the operational and NN-based retrieval at 869 nm for the NOAA-21 orbit that initially measured the plume from the Shiveluch eruption. The NN retrieves aerosol extinction profiles that are consistent with the operational retrieval algorithm, both for background stratospheric aerosol as well as Shiveluch's plume. The Ruang

eruptions have been the most significant events measured by NOAA-21's OMPS LP to date and thus represent an ideal test case to ensure our SNPP-trained NNs are applicable to major events measured by NOAA-21. Figure 8 shows the retrieved aerosol extinction coefficient at 997 nm for the operational and NN algorithms for the NOAA-21 orbit that measured the plume on April 30 from Ruang's eruption on the prior day. The inferred plume height (∼23 km) and retrieved extinction values agree well throughout the orbit.

Despite that calibration of the NOAA-21 data has not been finalized yet, these results suggest that the methodology presented here can produce NNs that are applicable to other OMPS LP instruments beyond SNPP without the need to re-train the models.

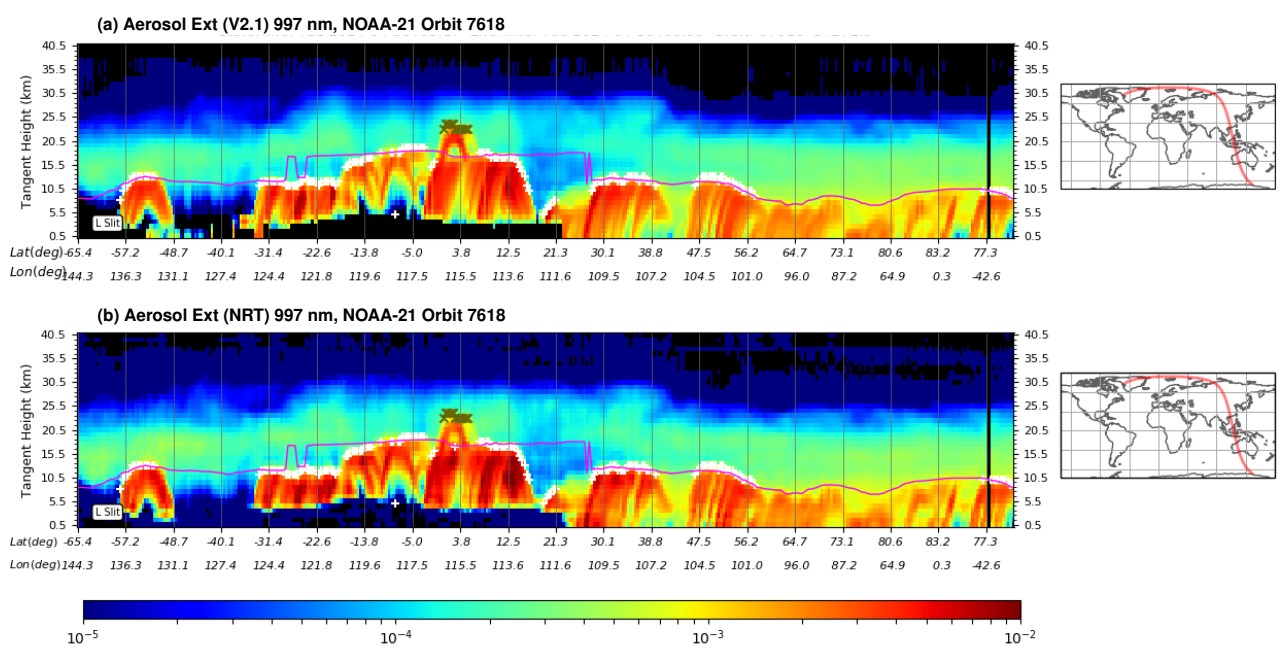

**Figure 8.** (a) NOAA-21 OMPS LP retrieved aerosol extinction at 997 nm for the left slit of orbit 7618 on 30 April 2024, which measured the plume from Ruang's second major eruption that month over Borneo. (b) The same but using the SNPP NRT algorithm. Both algorithms found that the plume reached a height of approximately 23 km, with similar magnitudes for the extinction throughout the plume as well as background aerosol throughout the orbit.



For cases where there are future iterations of the same instrument, our methodology can circumvent the limiting requirement of a long, diverse record to train generalized NNs by utilizing measurements from the older instrument.

## 4 Conclusions

We presented a near-real-time (NRT) aerosol retrieval product for OMPS LP based on neural networks (NNs). We found good agreement with OMPS LP's operational version 2.1 (V2.1) aerosol retrieval product, achieving zonal-mean errors that are generally within 20% at altitudes with sensitivity and non-negligible aerosol extinction. The NRT product exhibits the smallest biases in zonal means at 997 nm, with 869 nm providing similar accuracy but with slightly higher biases at low altitudes in the southern hemisphere. Given that Taha et al. (2021) recommend focusing on 869 and 997 nm for scientific studies using the

operational product, we similarly recommend that users focus on these wavelengths when using the NRT product, particularly 997 nm given its more consistent performance.

The NRT approach is $\sim 60\times$ faster than V2.1, requiring roughly 2 minutes to retrieve the extinction for an entire orbit. More quickly providing information on volcanic plume height after an eruption can be especially useful for groups like the Volcanic Ash Advisory Centers to inform aviation flight paths as well as providing advance notice to coordinate ground-based or in-situ

follow-up measurements just after large eruption or pyroCb events. With around 97% of the runtime spent loading the NNs into memory, this speedup factor can grow by more than an order of magnitude when processing a range of orbits. We found it necessary to train separate NNs for the southern hemisphere/short wavelengths and northern hemisphere/long wavelengths in order to achieve more consistent performance at all wavelengths and latitudes. If a single NN could reliably perform in both hemispheres, the factor speedup would double; future work should explore custom loss functions that can achieve this.

Our approach has two types of limitations: those related to the aerosol retrieval code, and those related to the NN approach. The limitations associated with the aerosol retrieval code inherently cannot be circumvented by our methodology; the NNs similarly struggle in situations where the retrieval code struggles to provide a consistent result or where it cannot retrieve the extinction (e.g., when the aerosol plume reaches the normalization altitude; Taha et al., 2022). The NN-specific limitations are associated with the training data. In general, these NNs cannot be reliably applied to other instruments or different versions

of radiances/extinctions without retraining; future work should explore applications of transfer learning to reduce the burden of updating the models as new versions of radiances/extinctions are developed. While applying NNs to data beyond those encountered during training can produce poor results, we found sufficient accuracy when applied to the HTHH eruption, which was more extreme than any events included in the training data set. The NRT approach also accurately identified the plumes for the recent Shiveluch and Ruang eruptions, which occurred outside of the temporal range covered by the training data. Our

results indicate that the NNs properly generalized for the problem and can be applied to future events measured by OMPS LP.

We also found that the NNs trained on SNPP data are consistent with the operational retrieval algorithm when applied to data from NOAA-21's OMPS LP without retraining. With multiple successors to SNPP and NOAA-21 planned over the next decade, future work should explore this in more detail, as it offers a way to circumvent the methodology's requirement of an extensive data record to apply our NRT approach.

*Code and data availability.* The MARGE software is available on GitHub at https://github.com/exosports/MARGE. All MARGE-related data and results for this work are publicly available at https://doi.org/10.5281/zenodo.11477425 (Himes et al., 2024a). The SNPP OMPS LP version 2.6 L1G data product is available at https://omps.gesdisc.eosdis.nasa.gov/data/SNPP_OMPS_Level1B/OMPS_NPP_LP_L1G_EV.2.6/. The SNPP OMPS LP version 2.1 L2-AER daily data product is available at https://snpp-omps.gesdisc.eosdis.nasa.gov/data/SNPP_OMPS_Level2/OMPS_NPP_LP_L2_AER_DAILY.2/. The the NRT aerosol data product's 7 most recent days are available at https://omisips1.

omisips.eosdis.nasa.gov/outgoing/OMPS/LANCE/LP-NRT-AER/. Imagery for the NRT product's retrieved aerosol extinction at 997 nm are available at https://ozoneaq.gsfc.nasa.gov/data/aerosols/#tab=image&prods=158&view=1.

*Video supplement.* An animation of the V2.1 and NRT average retrieved extinction coefficient between 19.5–21.5 km at 997 nm for the 2024 Ruang eruptions has been made available under a Creative Commons Attribution Non Commercial No Derivatives 2.0 Generic license at https://doi.org/10.5281/zenodo.11641540 (Himes et al., 2024b).

*Author contributions.* MDH was responsible for curating the aerosol data set used for the NNs, training the NNs, developing the NRT retrieval software, and writing the initial draft of the paper. GT provided guidance on the development of the NRT data product, developed the cloud height/type algorithm used in the NRT data product, and produced plots comparing the NRT data product with the V2.1 data product. DK set up the Python environment used by the NRT software, assisted with the release of the NRT data product, and led the data processing of NOAA-21 orbit 2153 from Level 0 to Level 2. TZ assisted in the development of the NRT software and data product. NAK

provided guidance on the development of the NRT data product. GT and NAK reviewed the manuscript and provided advice on the text and figures.

*Competing interests.* At least one of the authors is a member of the editorial board of Atmospheric Measurement Techniques.

*Acknowledgements.* The authors would like to thank the OMPS LP characterization team led by Glen Jaross for producing the Level-1 gridded data used in this work, the OMPS ozone SIPS team led by Matt Deland and Colin Seftor for Level-2 data production, Robert

Loughman and the OMPS aerosol science team, all other members of the OMPS Team for their work supporting the OMPS mission, Otmar Olsina for making the NRT imagery available on the NASA Ozone and Air Quality website, and James Johnson for feedback on the data product. We also thank contributors to NumPy, SciPy, Matplotlib, TensorFlow, Keras, Optuna, Dask, the Python Programming Language, and the free and open-source community. The data used in this effort were acquired as part of the activities of NASA's Science Mission Directorate, and are archived and distributed by the Goddard Earth Sciences (GES) Data and Information Services Center (DISC).





*Financial support.* MDH was supported by an appointment to the NASA Postdoctoral Program at the NASA Goddard Space Flight Center, administered by Oak Ridge Associated Universities under contract with NASA. GT was supported by NASA Earth Science SNPPSP Grant 80NSSC23K1037.



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
