# Peer review of "Using neural networks for near-real-time aerosol retrievals from OMPS Limb Profiler measurements"

_EGUsphere, 2024_

## Author Comment (AC1)

We thank the reviewer for taking the time to assess the manuscript and provide thoughtful, constructive comments. Below, the reviewer's comments are italicized in navy blue followed by our response to each point.

> *Line 1: „Among aerosol characterization methods"*
>
> *This is only a minor point, but why "characterization"? Does "characterization" include simulations and observations? I've never seen "characterization" in this context and its meaning should perhaps be explained?*
>
> *Line 2: "characterization"*

To characterize something is to describe its qualities. This term is more encompassing than simply saying "remote sensing" or "retrievals", given that aerosols can also be characterized via in situ measurements. We have replaced the second instance of 'characterization' with 'retrieval' to vary the verbiage.

> *Line 3: "but the radiative transfer modeling of scattering processes performed by traditional retrieval methods are too computationally expensive for near-real-time applications without simplifying assumptions."*
>
> *I'm not sure if this is really true. It will depend on the instrument, the number of limb measurements per orbit, the computational resources available. I'm aware of algorithms that can process an orbit in, e.g. an hour, making NRT retrievals possible, of course depending on the specific meaning of "NRT".*

This is a good point. It may not be true for all instruments, but it is true for the OMPS LP instrument + the available computing resourcing that we are using in this study. We have revised the text to better capture this:

> "… but the radiative transfer modeling of scattering processes performed by traditional retrieval methods can be too computationally expensive for near-real-time applications without simplifying assumptions, depending on the instrument and available computational resources."

> *Line 16: "cooler temperatures" -> "lower temperatures" ?*

Thank you for the suggestion. We felt that saying "lower temperatures … lower troposphere" was redundant, so we elected to vary the verbiage.

*Line 35: "The Ozone Mapping and Profiler Suite (OMPS) Limb …"*

*This sentence is incomplete.*

We appreciate the reviewer's suggestion, but we respectfully disagree with this comment. While it was grammatically correct before with a subject, verb, and objects, upon rereading it we feel it is a bit awkwardly worded which likely led to this comment. Consequently, we have changed the past tense to the past participle and omitted 'first' ("first launched" -> "was launched") which we hope improves readability.

*Line 41: "Gauss-Siedel" -> "Gauss-Seidel"*

Thank you, we have corrected this typo.

*Line 46: „their runtime is prohibitive to NRT applications unless compromises are made"*

*How long does it take to process one orbit with the method of Taha et al. (2022)?*

That algorithm requires around 2 hours to process an orbit on our processing system. In line with the above recommended change to line 3, we have modified this text:
"… their runtime can be prohibitive to NRT applications unless compromises are made …"

We have also added additional text in Section 1 that includes the aforementioned ~2 hour processing time for all three slits and six wavelengths:
"… the available computational resources result in just over 2 hours to process 1 SNPP orbit (and more than double that for NOAA-21 orbits), not including the time to downlink the data and process it into the Level 1 Gridded (L1G) radiance product."

*Line 52: "that produced that connect inputs and outputs"*

*Something seems to be missing/wrong here?*

Thank you for catching this typo, we have adjusted the text to correct this: "… the underlying physical processes that connect the inputs and outputs."

*Table 1, caption: "Date ranges of OMPS LP data considered."*

*Considered for training, validating, testing?*

*Line 67: "We utilize measurements during specific periods between October 2013 and December 2022 (Table 1)."*

*It is unclear, whether these periods are used for training the NNs? Or testing?*

Those data are used for both training, validation (monitoring training for overfitting) and testing the generalization of the model. Note that We have added text to clarify that these points. The caption is now "Date ranges of OMPS LP data considered for the machine learning data set", and the table footnote now includes "We select 10% of the above dates for our machine learning data set, totaling 241 days, which is subsequently split into training, validation, and test sets (see Section 2.2)."

*Line 76: "calculated from quantities available in L1G"*

*Please explain "L1G"*

L1G is the Level 1 gridded radiance data product. We have added text to define this at its first usage, which now occurs in Section 1:

"… not including the time to downlink the data and process it into the Level 1 Gridded (L1G) radiance product."

*Lines 71 – 78: what about O3? Isn't it necessary to consider O3 in some way? The Chappuis bands will have a significant effect on the shape of the LR profiles in the visible part of the spectrum.*

You are correct that O3 plays a role here. It primarily affects the shorter wavelengths (510, 600, 675 nm), but it negligibly affects the longest wavelengths (869, 997 nm) which are the recommended wavelengths of our NRT product.

The standard algorithm for OMPS LP's aerosol extinction product uses O3 climatology with an empirically determined corrective factor (Loughman et al., 2018). Section 5.4 of Loughman et al. (2018) investigated aerosol extinction changes due to this corrective factor and found that the corrective factor can induce up to a 20% change in the retrieved aerosol extinction at 675 nm. Note that the training data set uses the ozone-corrected aerosol retrievals.

However, this corrective factor is not saved out, and thus we must either (1) re-run all retrievals to obtain the exact O3 profile used in the retrieval, (2) use OMPS LP's standard O3 product, which is not consistent with what is used in the aerosol retrieval, or (3) omit O3 from the NN's inputs. #1 is not feasible due to the required computational resources to reprocess the full LP record, and #2 is not ideal given that it presents an inconsistency that is unlikely to be uniform across the OMPS LP record. Thus, we elected for #3. Rather than explicitly include the assumed O3 profiles among the inputs, our approach assumes that O3 is implicitly accounted for by the

NN's weights and biases given that the model sees O3-sensitive wavelengths (510, 600, 675 nm) and O3-insensitive wavelengths (869 and 997 nm), and that the shorter wavelengths' aerosol profiles were already corrected for the ozone absorption in the standard retrieval.

Consequently, we leave the assessment of the impact of including O3 profiles among the NN inputs to future work, and in the Conclusions we suggest this as an avenue for future inquiry:
> "Additionally, future work should consider how including ozone profiles among the NNs' inputs impacts the resulting accuracy, particularly at the shorter wavelengths considered in this study which are sensitive to ozone."

*Line 80: "These inputs correspond to the aerosol extinction coefficient reported in the OMPS LP aerosol retrieval version 2.1 data product"*

*Context is unclear? What does "These inputs" refer to? The extinction coefficients should be outputs (of the NN), right?*

"These inputs" refers to the bulleted list on lines 72-78.  Those inputs are paired with the extinction profile (the outputs of the NN).  We have adjusted the text to more clearly convey this point:
> "Each case within the data set is comprised of the above listed inputs paired with the corresponding aerosol extinction coefficient reported in the OMPS LP aerosol retrieval version 2.1 data product (V2.1; Taha et al, 2022)."

*Line 83: "we assume a value of $10^{-8}$"*

*Unit is missing (1/km)*

Thank you, we have added that into the text here and at the other placed mentioned later in the review.

*Line 85: "NN to be less than $10^{-8}$ are replaced with a fill value of -999"*

*Again, the unit is missing. What happens if the predicted value is e.g. $1.1 \times 10^{-8}$?*

A value of 1.1e-8 would remain as that value, consistent with the OMPS LP standard product. The standard product treats 1e-8 as no retrieved extinction, and since the NN is learning to emulate that algorithm, we adopt the same convention here.

*Line 92: "These correction methods introduce differences in the retrieved aerosol extinction coefficient, and so our input-output pairs do not have a perfect one-to-one relation."*

*This does not seem to be ideal, because the input/output data sets used for the training are not consistent. Can you quantify the effect on the estimated/retrieved aerosol extinction coefficients?*

This is indeed not ideal, but we have found that it is not a concern.  As discussed in lines 205-209 of the original manuscript, we also applied our methodology to a perfect one-to-one relation and found statistically similar results, that is, this assumption does not introduce any additional uncertainties beyond those inherent to the methodology.  This confirms our original hypothesis that the NN will learn to account for these differences in the input radiances and thus it shows that this inconsistency does not inhibit the application of the methodology.

*Line 99: "These data are split into training, validation, and test sets."*

*I'm not sure, how this relates to the periods listed in Table 1? Are you splitting the periods listed in Table 1? In the caption of table 1 you mention that 10% of the data are selected (for what purpose?). I'm not sure, how this fits to the 70%, 20%, 10% splitting mentioned in line 101. Please remove the inconsistencies.*

Among the date ranges in Table 1, we select 10% of those dates (241 days), as described in the table footnote.  Among those 241 days, we then split it such that 70% (169 days) are used for training, 20% (48 days) are used for validation, and 10% (24 days) are used for testing.  We have added additional text here to better clarify this:
"The aforementioned 241 days of data are split into training, validation, and test sets."

*Line 122: "as we found poor performance at 510, 600, and 675 nm in the southern hemisphere,"*

*I'm not too surprised by this finding, because O3 will affect this spectral region and the SH experiences very low O3  concentrations at higher latitudes during the O3 hole season.*

*Line 146: "The notable exceptions to this are lower altitudes in the southern hemisphere at the shorter wavelengths"*
*See previous comment.*

This behavior is consistent with prior work by Taha et al. (2021) and is due to the weak aerosol backscattering signal, as discussed in this manuscript and in Taha et al. (2021).  Shorter wavelengths experience a stronger Rayleigh scattering signal, which dwarfs the aerosol

backscattering signal at the extreme single-scattering angles of the far southern latitudes. This is why the behavior is most pronounced at 510 nm, where Rayleigh scattering is the strongest among the 6 considered wavelengths. O3 certainly plays a role here (see the earlier comment on this point), but our results suggest that Rayleigh scattering has a larger impact than O3 at the 3 shortest wavelengths considered in this study. In particular, Fig. 1 shows that we achieve better agreement at 600 nm than at 510 nm, which is the opposite of what would be expected when considering O3 cross sections as they are greater at 600 nm than 510 nm. If this behavior were primarily attributable to O3, then Fig. 1 should show worse agreement at 600 nm. The NN retrieval struggles where the standard retrieval accuracy is limited, which is the case for the shorter wavelengths at large scattering angles (see Taha et al., 2021).

*Figure 1: It would be interesting to see the differences in concentrations as a function of time, particularly in the SH at high latitudes. I assume that during the O3 hole season the differences can be significantly larger than in the temporal mean. Please show a time-altitude-contour plot of aerosol extinction differences for different latitudes.*

Thank you for the suggestion. We produced the mentioned plots for 5-degree zonal means, and we do not see evidence of significant deviations from the temporal mean during the O3 hole season. During that time of year, the aerosol extinction is very low inside the vortex, and our NRT method generally agrees with the standard product. The only apparent correlation we see is that the NN tends to predict more extreme values than the standard product, that is, positive biases are generally associated with larger extinction values while negative biases are generally associated with smaller extinction values. See Figure R1 below for one of these plots. We have added some new text to Section 3.1 to address this point:

> "We similarly see larger percent errors during the southern polar vortex, which is attributable to the small aerosol extinction within the vortex; outside of the vortex, we find that biases are generally consistent with the temporal mean."

[Figure]

**Figure R1.** Daily zonal mean aerosol extinction coefficient between 80-85°S latitude. Top: OMPS LP V2.1 product. Middle: Like the top panel, but for the near-real-time product. Bottom: Differences between the top and middle panels in percent. Periods where the aerosol extinction is near 0 at altitudes of 18-20 km coincide with the southern polar vortex and seasonal thinning of the ozone layer over Antarctica. Biases in this period are generally consistent with the temporal mean reported in the manuscript, except where aerosol extinction is negligible.

*Caption Fig. 1: "The dashed line is the tropopause altitude"*

*Where does the tropopause altitude data come from?*

These data are zonal averages of the GEOS FPIT data. We have updated the text accordingly:
"The dashed line is the zonal mean GEOS FP-IT tropopause altitude."

*Fig. 2 and Fig. 3: "Le Soufriere" -> "La Soufriere"*

Thank you for catching this typo, we have corrected it.

*Fig. 2: Please explain „WC"*

We have expanded this to "USA west coast fires" for clarity.

*Fig. 2: How is the sAOD determined, i.e. what altitude range/tropopause data?*

The tropopause data is from GEOS FP-IT, as mentioned in the comment above.  The altitude range is from that tropopause altitude to 37.5 km.  We have added text to clarify this point:
"… by integrating the aerosol extinction from the tropopause altitude to 37.5 km."

*Fig. 4: "Tangent height" should read "altitude" or "height", right?*

"Tangent height" refers to the altitude at the tangent point observed by OMPS LP.

*Fig. 4: Unit is missing (1/km)*

Thank you, fixed.

*Caption Fig. 3: "Ha'api" -> "Ha'apai"*

Thank you for catching this typo.  We have updated the text in line with the latest recommendations to refer to this event as the "Hunga eruption", and so all mentions of Ha'apai have been removed from the text.

*Line 185: "hardware, achieving a ~60x speedup compared to V2.1."*

*OK, so a full orbit takes about 120 minutes with the V2.1 processor, compared to 90 minutes orbit duration. This means that with some adjustments it would be possible to process one orbit in 90 minutes with the "full physics" version.  Also: what machine are you using for the calculations?*

Processing the standard algorithm in 90 minutes for a single orbit would not meet NASA's definition for NRT (within 3 hours of the measurements) due to the time involved in downlinking the data and processing it through to the L1G product.  Furthermore, that would

still leave a difference of 88 minutes between our NN-based approach and this hypothetical speedup to the standard algorithm, and speeding up the physics-based algorithm to meet NRT requirements would require either (a) significant limitations (reduced number of wavelengths and slits) or (b) a significant investment of money and worker hours to enhance the retrieval algorithm and the processing capabilities of our system.  A major benefit of our approach is that it minimizes the delay between when we acquire the measurements and when we release the results to just a few minutes.

We are using NASA's Atmospheric Composition Processing System to run both the standard and NRT algorithms, which is now included in Section 1:

> "In the case of the NASA Atmospheric Composition Processing System used to produce the NASA OMPS LP aerosol product …"

> *Line 190: „When experimenting with changes to the radiative transfer-based aerosol retrieval algorithm, our methodology can therefore significantly reduce the computational resources required to determine how such changes would affect the mission's complete record."*
>
> *I'm not sure this would really be the case. If you have differences of 20% and more between the NRT and the full physics data set, how would that help to test how changes would affect the entire record? Perhaps I'm missing the point here?*

It depends on the specific goal of these hypothetical changes.  If it is important to consider the impact of such changes on the full OMPS LP record and if the expected changes do not require a precision <20%, our methodology can significantly reduce the computational resources necessary to generally evaluate that.  However, in practice, such changes can typically be investigated using select orbits or periods of time (e.g., a month), in which case our methodology is not helpful.  Consequently, we have removed the quoted text, as it is at best a niche application of the methodology.

> *Fig. 5: "Tangent height" should read "altitude" or "height", and the unit is missing (1/km).*

Done.  See the responses above.

> *Fig. 6/video supplement: The unit (1/km) is also missing in the video supplement.*

Done.

*Line 195: "with strong biases in the southern hemisphere and shorter wavelengths"*

*Did this occur in all seasons? Again, I presume that O3 and the O3 hole play an important role here.*

To some extent, yes.  This bias is generally correlated with the scattering angle's seasonal variation, which is observed in the southern hemisphere even at 869 and 997 nm (see Fig. 1) and are negligibly affected by O3.  It is due to the weak aerosol backscattering signal limiting the precision of the retrieval.  Since the aerosol extinction is typically minimal at high latitudes, the percentage differences are large even though the absolute differences are small.

*Line 196: "as well as OMPS LP's Sun-synchronous orbit"*

*You mean the large scattering angles in the SH and the small ones in the NH? This is not only a consequence of the sun-synch orbit, but also of the viewing direction. OMPS-LP could also be viewing in the opposite direction. Then you would have small/large scattering angles in the SH/NH.*

Yes.  We have added text to explicitly mention this factor:
> "… as well as OMPS LP's Sun-synchronous orbit and viewing direction."

*Line 207: "This confirms our approach's implicit assumption that the NNs can learn to handle the minor differences in corrections applied to the radiances between versions 2.5 and 2.6,"*

*I'm not sure that this implies that the NN can handle the differences between versions 2.5 and 2.6. That would be surprising, right? It probably means that the differences were not so large!?*

*But it would be very interesting to mention the differences in performances in a quantitative way.*

The results from the NNs trained on v2.5 and v2.6 are statistically the same, so yes, it does show that the NN can handle the differences when predicting the retrieved extinction profiles.  If it couldn't, then we would see statistical differences.  The NN is learning an approximation to map the inputs to the target outputs, and since the differences between versions 2.5 and 2.6 are consistent throughout the data set (consider the tangent height and stray light correction algorithms to be transformations applied to the same underlying data), they can be implicitly accounted for via the NN's weights and biases.

Note however that if one applies the v2.6 model to v2.5 radiances (or vice versa), then the differences are significant. This is not surprising, given that those radiances are out of the training data distribution.

Since it is already stated the results of NNs trained on both versions of radiances are statistically the same, we do not feel it is useful to repeat the already quoted results.

> *Line 216: "In the former case"*
>
> *Not entirely clear what "former" refers to? High altitudes or SH? or both?*

That refers to high altitudes and the SH. To better clarify this, we have replaced the quoted text with, "For high altitudes and in the southern hemisphere".

> *Fig. 7: "Tangent height" should read "altitude" or "height", and the unit is missing (1/km).*
>
> *Fig. 8: "Tangent height" should read "altitude" or "height", and the unit is missing (1/km).*

See the relevant comments above.

> *Conclusions: perhaps one could mention as an outlook that the approach could be extended to consider O3 as well? This would probably improve the performance of the NRT data product.*

Thank you for the suggestion, we have added text about this in the Conclusions discussing this as a potential future investigation (see earlier comment for a quote of the added text).

> *Line 280: "that the NNs properly" -> "that the NNs are properly"?*

Thank you for the suggestion. We have omitted "properly" for conciseness and clarity.

---

## Author Comment (AC2)

We thank the reviewer for taking the time to assess the manuscript and provide thoughtful, constructive comments.  Below, the reviewer's comments are italicized in navy blue text followed by our response to each point.

*As someone without experience in neural networks, I found the description in section 2.2 somewhat difficult to follow. This is probably standard in ML literature, but given the atmospheric journal/audience I think some context would be helpful here. For example, values for the various activation functions, epochs, batch sizes, cyclical and minimum learning rates are nicely provided, but what these terms mean and how these choices impact the retrieval are not clear except for a brief mention of overfitting. I don't mean to turn this into a primer on ML, but a bit more of a link between these parameters and the results would be nice.*

Indeed, it is standard for ML literature, but it is a great point that the audience for this is not ML experts but rather atmospheric scientists.  We have added additional information in Sec. 2.2 that we hope better clarifies these terms and their importance for the retrieval methodology. Excerpts of these additions are provided below:

"These parameters [layer types and number of nodes per layer] are related to the overall complexity that can be captured by the NN."
"... activation functions, which introduce non-linearities into the model such that it can approximate complex behaviors."
"... trained for 60 epochs (number of iterations over the data set; relates to model convergence) using the Adam optimizer (controls how the NN weights and biases are updated), a batch size of 256 (number of samples considered in each training iteration; relates to variance in the gradient and thereby how the model learns), and a cyclical learning rate (scaling factor for magnitude of model updates; cycling reduces number of epochs needed to train model to a certain performance, e.g., Smith, 2015; Himes et al., 2022) ..."

*A lot of emphasis is given to the computational cost of v2.1 and the inability to produce a near real time product using this approach. However, I don't think these conclusions are justified by the paper in its current form. For example:*

*Line 4: "processes performed by traditional retrieval methods are too computationally expensive for near-real-time applications without simplifying assumptions."*

*OMPS retrievals as implemented in v2.1 are an "embarrassingly parallel" problem across profiles, slits and wavelengths and seemingly could be sped up if needed. This may be cost prohibitive, but the paper does not discuss what hardware is currently required to process v2.1 or potential increases needed for a real time v2.1 algorithm vs NN. As it is,*

*the OMPS-LP v2.1 data is processed at one day per day, so it seems the throughput of the current v2.1 system is not a problem, only potentially the lag.*

*Similarly, the cost of running the v2.1 algorithm and NN algorithms is discussed (line 185), but it's unclear what hardware was used for the comparison. Is the NRT approach ran on the same hardware as the v2.1 retrieval? Or are there special hardware requirements for this NRT retrieval (GPUs?) It is mentioned that 97% of the time is spent loading the NN into memory, does this require a machine with a large amount of memory (or more than the v2.1 uses)? Is the training factored into this analysis of computational cost?*

Yes, it is an "embarrassingly parallel" problem.  Yet, computational resources are limited, and the design of our processing system introduces additional constraints.

Near-real-time processing for OMPS limb data takes place in an automated scheduling and processing system. The automated processing system is a key component to producing NRT data in an on-going basis. Furthermore, the implementation of the non-ML NRT algorithms are essentially the same as the ones used to produce the publicly released standard products (our NRT aerosol algorithm presented here is the exception) but without the requirement that the processing system wait for delayed packets and auxiliary pressure/temperature data.  To leverage the existing facilities for NRT processing, the limb granule size of one orbit per granule used in non-NRT processing is preserved.

The current retrieval algorithm has embarrassingly parallel characteristics in that each image in each slit can be retrieved independently and in parallel.  However, the processing system limits us to running an entire granule on only one processing node, i.e., we do not break up a granule over multiple nodes for NRT because we do not do this in nominal processing.  The processing nodes themselves consist of multi-core shared memory systems and so any parallelism is limited to farming the image-slit retrievals for the entire granule over only the cores on that node.  This places a fundamental limit on how much the algorithm can be parallelized when using this processing system.  This may change in the future, but it would require a significant investment of both money and worker-hours.

It might be concluded that the inadequacy of the physics-based algorithms for NRT processing will be overcome as nodes with more CPUs become available to process data within the NRT time constraint.  However, there are developments that work against that.  Experience with Suomi-NPP allowed the sample time for the follow-on mission, NOAA-21 OMPS LP, to be more than halved while maintaining sufficient signal-to-noise (i.e., the number of images per granule is more than doubled as is the concomitant retrieval processing time).  Another is that improvements to the physics-based model can reduce the speed of model, potentially breaking compliance with NRT requirements.  Perhaps more importantly when considering future developments, our current retrieval model is a 1D approximation, and 2D tomographic retrievals that leverage information from measurements before/after the event of interest add additional computational costs.  For these reasons, it seems likely that as computational

resources increase, so too will the computational complexity of the retrieval algorithm. Our NN-based methodology ensures that we continue to meet the NRT requirements regardless of how the computational resources and retrieval algorithm change over time.

The "60x faster" result we present is determined by comparing the average processing time (wall clock time) for a single orbit when using the same processing system, and thus it represents an apples-to-apples comparison. The NRT approach requires less RAM (by a factor of ~2) and less CPU time (by orders of magnitude) and so represents a more efficient algorithm in terms of computational resource requirements.

Training time is not factored into the "60x faster" metric reported. It requires around 13 hours to train each model using an NVIDIA V100 GPU, or just over 1 day in total. In the context of reprocessing even 1 year of OMPS LP data, this is a negligible amount of time. However, it is a good point that we did not address in the original manuscript.

We have added text into Section 1 to better clarify these points:
> "In the case of the NASA Atmospheric Composition Processing System used to produce the NASA OMPS LP aerosol product, the available computational resources result in just over 2 hours to process 1 SNPP orbit (and more than double that for NOAA-21 orbits), not including the time to downlink the data and process it into the Level 1 Gridded (L1G) radiance product. At present, this does not meet NASA's prevailing NRT definition of within 3 hours of the observations. While these runtimes can be reduced by newer computing hardware, processing speed improvements can be offset by updates to the realism of the radiative transfer and retrieval models (e.g., tomographic retrievals as in Zawada et al., 2018)."

> *Line 46-48: Is the time to process the retrieval of a single profile in v2.1 so long that it precludes NRT applications? I would have guessed (maybe incorrectly) the downlink, attitude solution, L1 calibration, atmospheric reanalysis etc. would have been a larger contributor to any lag in NRT products than the retrieval itself. How "NRT" could the NN version be in practice, given this is proposed as a major benefit of the proposed system?*

Yes, much of the NRT processing time is spent on the downlink and L1 calibration. This leaves a very limited amount of time for the aerosol retrievals to meet NASA's NRT definition (available within 3 hours of the measurements). Version 2.1 of the standard aerosol algorithm requires around 2 hours to process 1 orbit on our processing system. In an ideal world where we are not limited by computational resources, funding, and worker-hours, it could be sufficiently parallelized to meet that NRT definition, but this is not feasible for the reasons mentioned in the previous comment.

The NN-based algorithm requires around 2 minutes to process 1 orbit on ACPS, representing a ~2-hour speedup vs. the standard algorithm, and it ensures we meet NASA's NRT definition.

*Line 71: What are the outputs of these input-output pairs? Is it cloud top altitude, enhanced layer and PSC, as marked in Figure 4 as well as multi-wavelength extinction?*

The outputs are the multi-wavelength aerosol extinction profiles, as mentioned on lines 80-81 in the original manuscript. For cloud/enhanced aerosol/PSC altitude, we apply the standard aerosol algorithm's approach, as it is very fast and doesn't require the speed benefits of ML. We have revised the text to better clarify these points:

> "Each case within the data set is comprised of the above listed inputs paired with the corresponding aerosol extinction coefficient reported in the OMPS LP aerosol retrieval version 2.1 data product …"
> "To determine the altitude of clouds, enhanced aerosols, and polar stratospheric clouds, we utilize an updated version of the detection algorithm of V2.1, since it is already sufficiently fast and does not require further speed improvements from ML."

*Line 94: I would change to "no NASA OMPS LP aerosol retrieval version" as the University of Bremen and University of Saskatchewan OMPS-LP aerosol products both use version 2.6.*

Thank you for the suggestion, this is a good point. We have updated the text accordingly.

*Line 95: What is meant by "differences in correction methods are consistent"?*

Versions 2.5 and 2.6 of the gridded radiances product use different approaches for tangent height and stray light corrections. We can view those correction algorithms as some transformation function applied to the same underlying data. Since each version applies those corrections consistently throughout the product, and the differences between versions are consistent (even if not readily apparent to a person's eyes), the NN can implicitly learn how to account for those consistent differences. This is an important assumption for our methodology because, as discussed in that section, there is not yet a NASA OMPS LP aerosol retrieval version that uses version 2.6 of the gridded radiances product, and only the version 2.6 radiance product is produced in near real time, so we must use that version out of necessity. It is thus a requirement that the NN is able to account for the differences between those versions, and indeed our results show that the NN is able to do so. We have revised the text here to better clarify this point:

> "Since the differences in correction methods can be viewed as different transformation functions applied to the same underlying data, our methodology ignores them and assumes that the NN will learn to perform the transformation from version 2.6 radiances to V2.1 aerosol extinction coefficients."

*Line 125-130: Probably obvious for someone in the ML field, but how are results from these two NNs put together?*

This is a good question, as there are multiple ways this could be performed. In our case, we make predictions using each of the two NNs, then select the subset of predictions that are relevant for each NN, and finally combine them into a single array. This approach was chosen because it is not only simple algorithmically, but more importantly a given wavelength, latitude combination will only have a relevant prediction from one NN. We have added some additional text in Sec. 2.2 to better clarify how this is performed:

> "To retrieve on 1 orbit, predictions are made with both models, and then the aforementioned relevant subsets of predictions from each model are combined."

---

## Author Response (AR2)

To the editor,

Below is our response to the final comment from the anonymous reviewers.

> *One little problem remains, however: The y-axis label of Figs. 4, 5, 7 and 8 is still "Tangent height" and this is not correct in my opinion. "Tangent height" refers to the "observation height" (for lack of a better word) of the limb-scatter measurements. OMPS-LP measures limb radiances as a function of tangent height and from that you retrieve aerosol extinction as a function of "altitude" or "height". The actual vertical extinction profile is independent of the limb measurements, so why should its vertical coordinate be "tangent height"?*

We note that while the true vertical extinction profile is independent of the limb measurements, our retrieved vertical extinction profile is dependent on the limb measurements, specifically how they are processed from the Level 0 data to the Level 1 gridded radiance product, meaning our retrieved profile is ultimately a function of the assumed tangent heights of the limb measurements. Despite this, we have chosen to update the mentioned figures to say "Altitude" rather than "Tangent height" because it is a minor distinction and it brings these figures into consistency with the labeling used in Fig. 1.

> *Figs. 4, 5, 7, and 8 have been updated to show "Altitude" as the y-axis label, as requested.*